# Non-Palpable Breast Cancer: A Targeting Challenge–Comparison of Radio-Guided vs. Wire-Guided Localization Techniques

**DOI:** 10.3390/biomedicines12112466

**Published:** 2024-10-27

**Authors:** András Drozgyik, Dániel Kollár, Levente Dankházi, István Á. Harmati, Krisztina Szalay, Tamás F. Molnár

**Affiliations:** 1Department of Burns and Plastic Surgery, Petz Aladár University Teaching Hospital, 9024 Győr, Hungary; 2Doctoral School of Clinical Sciences, University of Pécs Medical School, 7624 Pécs, Hungary; tfmolnar@gmail.com; 3Kirurgkliniken, Värnamo Sjukhus, 331 56 Värnamo, Sweden; 4Department of Radiology, Petz Aladár University Teaching Hospital, 9024 Győr, Hungary; 5Department of Mathematics and Computational Sciences, Széchenyi István University, 9026 Győr, Hungary; harmati@sze.hu (I.Á.H.); szalayk@sze.hu (K.S.); 6Department of Operational Medicine, University of Pécs Medical School, 7624 Pécs, Hungary; 7Department of Thoracic Surgery, Petz Aladár University Teaching Hospital, 9024 Győr, Hungary

**Keywords:** breast cancer, ROLL, wire guided localization, sentinel lymph node marking

## Abstract

**Background:** The incidence of non-palpable breast cancer is increasing due to widespread screening and neo-adjuvant therapies. Among the available tumor localization techniques, radio-guided occult lesion localization (ROLL) has largely replaced wire-guided localization (WGL). The aim of this study was to compare the ROLL and WGL techniques in terms of the effectiveness of isotopic marking of axillary sentinel lymph nodes and to assess patient perspectives along with surgeon and radiologist preferences. **Methods:** A single-center, prospective, randomized study enrolled 110 patients with non-palpable breast lesions (56 ROLL, 54 WGL). Breast type, tumor volume, location, histological and radiological features, and localization/surgical duration were evaluated in the context of sentinel lymph node marking using isotope (technetium-99m-labeled human serum albumin) and blue dye. Statistical analysis was performed with significance set at *p* < 0.05 and strong significance at *p* < 0.01. **Results:** A single-center, prospective, randomized study enrolled 110 patients with non-palpable breast lesions (56 ROLL, 54 WGL). Breast type, tumor volume, location, histological and radiological features, and localization/surgical duration were evaluated in the context of sentinel lymph node marking using isotope (technetium-99m-labeled human serum albumin) and blue dye. Statistical analysis was performed with significance set at *p* < 0.05 and strong significance at *p* < 0.01. **Conclusions:** While ROLL provided advantages in terms of patient comfort and logistical simplicity, WGL was superior for axillary sentinel lymph node marking, particularly in inner quadrant tumors, suggesting that WGL may be preferred in these cases.

## 1. Introduction

It is estimated that approximately one in eight women will develop invasive breast cancer at some point during their lifetime. In 2020, breast cancer was the most frequently diagnosed cancer among women, representing 11.7% of all cancers [1]. Advances in screening and effective neoadjuvant treatments (NAT) have increased the detection of non-palpable breast tumors, necessitating accurate preoperative localization to ensure that resections are carried out in a timely and safe manner [2,3,4].

Wire-guided localization (WGL), introduced in 1965, was the first method for localizing non-palpable breast tumors [5]. While WGL is cost-effective and widely accessible, it has several limitations, including difficulty in wire placement and the potential for excessive excision of healthy tissue [6,7]. Despite these drawbacks, WGL remains in use, particularly when sentinel lymph node (SLN) marking is not required or when multiple isotope markings are necessary.

Radio-guided occult lesion localization (ROLL), introduced in 1998 [8], uses radioactive colloid for tumor and same session SLN marking [9]. ROLL reduces excision volume, improves cosmetic results, and enhances lesion centricity [6,10,11].

Despite these advantages, studies have shown no significant difference between ROLL and WGL in successfully localizing non-palpable breast lesions [12]. However, a key methodological difference exists in SLN marking: ROLL does not use peri-areolar isotope injection; instead, the sentinel node is marked with a single isotope injection at the tumor site within the breast. To date, no randomized studies have directly compared the effectiveness of these two classic methods in the context of SLN marking.

In the face of these challenges, the role of NAT is becoming increasingly important as it not only targets the disease but also enhances surgical options. NAT aims to shrink tumors prior to surgery and reduce the likelihood of cancer spreading to the axillary lymph nodes. This may facilitate less invasive surgical approaches such as sentinel lymph node biopsy (SLNB) and targeted axillary dissection, reducing the need for axillary lymph node dissection, which carries a higher risk of complications [13]. However, it is important to note that NAT may also have drawbacks: it can damage lymphatic vessels, and there may be a varying degree of response in the axillary lymph nodes, increasing the false negative rate for sentinel biopsy and thus decreasing the reliability of sentinel node marking, a factor that should be considered [14].

Finding the right place for new, emerging technologies such as magnetic, radioactive, radar, or radiofrequency location methods [12] requires a scrupulous evaluation of our own current practice with the most affordable modalities in order to proceed to the next, more advanced stages.

### Aims

The primary objective of this study was to compare the efficacy of the ROLL and WGL techniques in achieving feasible isotopic marking of axillary SLN.

The secondary objective was to evaluate the accuracy, efficiency, and ergonomics of ROLL compared to WGL by examining radiological and pathological outcomes. Subjective factors such as patient pain perception and surgeon and radiologist preference were also assessed with questionnaires.

## 2. Materials and Methods

### 2.1. Study Design

This single-site, prospective, randomized, non-blinded comparative intervention study was conducted at our high-volume cancer-center university hospital. All surgical procedures were performed by the same surgeon (AD) who also attended personally all marking procedures.

### 2.2. Patient Selection and Randomization

A total of 110 female patients between the ages of 36 and 80 years with non-palpable breast lesions were included in this study. Patients were randomized into two groups: the ROLL group (*n* = 56) and the WGL group (*n* = 54). Randomization was carried out using a random number generator after obtaining informed consent (and basic anthropometric data) from all participants. The ratio of mammography screening for primary tumor detection and complete clinical regression to NAT ratio among the ROLL and WGL groups was 43/13 vs. 44/10, respectively shown in Table 1.

A total of 43 patients (77%) in the ROLL group, and 44 patients (79%) in the WGL group had a non-palpable tumor due to early detection with cancer screening. NAT resulted in regression of the primary tumor to non-palpable size in 13 patients (23%) in the ROLL group and in 10 patients (18%) in the WGL group. There is no significant difference between the two groups in cancer screening and NAT ratios (*p* = 0.5449).

### 2.3. Breast and Tumor Characterization

Breast size was classified based on brassiere size into three categories: A (small), B and C (medium), and D or larger (large). Breast tissue type was determined by a radiologist using mammography images and categorized into five distinct types: glandular (1), adipose (2), retro-mamillary fibrosis (3), adenotic (4), and fibrotic (5) (Table 2 and Table 3, and Appendix A).

The distance of the tumor from anatomical landmarks such as the nipple, skin surface, and pectoral fascia was measured using ultrasound. Tumor localization within the breast was documented in terms of four quadrants—upper-inner, upper-outer, lower-inner, and lower-outer—as well as the central area. Tumor localization distribution is shown in Table 4.

All patients underwent a preoperative triple assessment including physical examination, mammography, breast ultrasonography and ultrasound-guided core biopsies. This approach allowed classification of lesions according to the Breast Imaging-Reporting and Data System (Bi-RADS) and facilitated the subsequent histopathological diagnosis.

### 2.4. Tumor Marking Procedures

Ultrasound-guided tumor marking was performed in 50 cases (93%) of the WGL group and 52 cases (93%) of the ROLL group. For four patients from each group (7%), stereotactic guidance was used, based on the tumor’s radiological characteristics. The duration and difficulty of the marking procedures were recorded by the radiologist and rated on a Visual Analog Scale (VAS) from 1 (easy) to 5 (difficult) [15]. In the WGL group, tumor marking was followed by surgery on the same day, while in the ROLL group, surgery was performed the following day, 16–32 h after isotope injection.

### 2.5. Lymph Node Marking

All patients underwent dual lymphatic pathway marking. This included the isotope injection 16–32 h before surgery and patent blue dye injection approximately 10 min before SLN surgery.

### 2.6. Procedural Assessment

The procedural assessment involved gathering feedback from surgeons, radiologists, and patients. Surgeons completed a standardized questionnaire documenting patient demographics, duration of localization procedures, and perceived difficulty of lesion localization during excision, rated on a VAS scale of 1 (very easy) to 5 (very difficult). Radiologists also filled out a questionnaire evaluating the complexity of marking non-palpable tumors. Patients were asked to rate their pain during the marking procedure on a VAS scale of 0 (no pain) to 10 (worst possible pain).

### 2.7. Post-Marking Procedures

Following the preoperative marking, resection of the primary tumor and axillary SLN was performed by the same surgeon in all patients, regardless of whether ROLL or WGL was used. The resected specimens were prepared for pathological examination using the Specimen Plate system as described in our previous publication [16]. This system, along with precise orientation techniques, allowed for accurate in situ localization of the tumor and surrounding tissues, ensuring comprehensive pathological evaluation. Additionally, the weight of the resected specimens was recorded.

### 2.8. Histopathological and Imaging Analysis

Mammography images and histopathological findings were analyzed to determine the adequacy of the resection to ensure radiologically and pathologically clear margins. Data on these outcomes were systematically recorded and evaluated.

### 2.9. Statistical Analysis

Statistical analysis was performed using Python 3.11.5 with packages such as Pandas 2.0.3, SciPy 1.11.1, Seaborn 0.12.2, and Matplotlib 3.7.2. We present the *p*-values rounded to four decimal places.

The Mann–Whitney test was used to evaluate differences between ROLL versus WGL. A *p*-value of <0.05 was considered statistically significant, strong statistical significance was proven if *p* < 0.01.

The mean age of patients was 62.4 ± 11.31 years in the ROLL group and 59.56 ± 10.1 years in the WGL group, with 95% confidence intervals of (59.37, 65.43) and (56.8, 62.32), respectively. The mean BMI was 26.24 ± 3.61 kg/m^2^ in the ROLL group and 25.89 ± 3.19 kg/m^2^ in the WGL group, with 95% confidence intervals of (25.27, 27.2) and (25.01, 26.76), respectively. No significant differences were found in age or BMI distribution between the groups.

The Kolmogorov–Smirnov test showed no significant differences in BMI (*p* = 0.9339) or age (*p* = 0.1177) distributions between the groups. Similar results were observed with the Mann–Whitney and Cramer–von Mises tests. One-way ANOVA also indicated no significant differences in mean BMI (*p* = 0.5932) or age (*p* = 0.1681) between the groups. (See statistical result figures in Appendix A).

## 3. Results

The difficulty of preoperative lesion marking from the radiologist’s perspective (VAS scale from 1 to 5) was 1.75 ± 0.96 in the ROLL group and 2.31 ± 0.97 in the WGL group, with a strong significant benefit of the ROLL technique (*p* = 0.0002).

Preoperative marking took 2.93 ± 3.71 min in the ROLL group and 3.92 ±3.16 min in the WGL group, showing strong significant superiority of the ROLL technique (*p* = 0.001).

Patients’ subjective pain (VAS 0–10) during preoperative marking also showed strong significant advantage of ROLL (1.41 ± 1.42) versus WGL (3.78 ± 2.03) *p* = 0.0000).

The ROLL technique is also significantly superior to the WGL technique in terms of the surgeon’s subjective perception of the operative difficulty of lesion localization (VAS scale 1.91 ± 0.96 versus 2.29 ± 1.04, respectively, *p* = 0.0197).

Skin-to-skin operation time was similar in the ROLL group (71.05 ± 18.92 min) and the WGL group (69.26 ± 14.51 min) (*p* = 0.9593).

Removed breast tissue weight (pathology specimen) was 88.48 ± 45.25 g for ROLL patients and 72.43 ± 33.55 g in the WGL group. Multiple statistical tests were carried out due to inconsistent significance. The Mann–Whitney (*p* = 0.019) and Cramer–von Mises (*p* = 0.0162) tests show significantly less resected tissue in WGL patients, while the conservative Kolmogorov–Smirnov test reached no significance (*p* = 0.0637).

All 110 patients underwent successful operations with clear and safe resection margins following either ROLL or WGL marking. The figures presenting the detailed results can be found in Appendix A.

### Sentinel Lymph Node Marking Results

We compared the percentage of SLN isotope marking failures across different tumor quadrants for both techniques (Table 5 and Table 6). The ROLL technique demonstrated a significantly higher failure rate for radioisotope SLN marking compared to WGL combined with peri-areolar radiotracer injection. Specifically, for tumors in the lower-inner quadrant, the failure rate for ROLL was four times higher (67%) than for WGL (14%) (*p* = 0.0265). Similarly, this increased failure rate was observed in the upper-inner quadrant as well (40% versus 10%, respectively, *p* = 0.0607).

*Study limitations*: Due to the small sample size, we cannot rule out the possibility that some differences may not be statistically significant.

Full details of the statistical comparisons, including *p*-values, are summarized in Table 7 and Table 8 and in Appendix A. Statistically significant differences are marked with *, and strong significance with **, with all significant results favoring WGL.

In the majority of cases, patients with non-detectable SLN using the ROLL technique were characterized by large breast size and fatty involution (Figure 1).

In the WGL group, SLN isotope identification failure was not as common as in the ROLL group. Even in large breasts with fatty involution (Figure 2).

Peri-areolar radioisotope injection associated with wire-guided marking had less negative impact on the axillary SLN marking, even in cases with large breast size and fatty involution.

The distribution of non-palpable tumors according to TNM (tumor size, node involvement, and metastasis) classification is shown in the following figure, based on their size and characteristics observed during histopathological processing. The distribution of tumor sizes is similar between the two groups examined (Figure 3 and Figure 4).

## 4. Discussion

Although non-palpable breast tumors carry a better prognosis, their treatment poses challenges on several fronts. First and foremost is the need for preoperative radiologically guided localization techniques and more intricate surgical procedures.

NAT can reduce tumor size and facilitate downstaging from cN1 to ycN0. However, despite this potential for downstaging, conversion from mastectomy to breast-conserving surgery remains underutilized in practice. Factors influencing this decision often include tumor location, size, and extent of disease at baseline [4].

In addition, the identification of sentinel lymph nodes (SLNs) following neoadjuvant treatment has been of concern due to reports of heterogeneous identification rates (IR) and increased false negative rates. Consequently, some patients undergo axillary lymph node dissection regardless of their response to NAT, leading to potential overtreatment [17]. Improving the reliability of SLN marking is essential, particularly for patients with ycN0 status, as successful identification of sentinel nodes can reduce unnecessary surgical interventions.

Therefore, it is crucial to improve the effectiveness of sentinel lymph node marking with each localization technique to optimize both surgical outcomes and patient quality of life.

NAT also decreases the chance of axillary SLN marking even after peri-areolar isotope injection [14,18,19,20]. In cases of tumor regression, where the tumor becomes non-palpable and a ROLL procedure is performed, there is an increased risk that the SLN may not be marked with isotope. This increases the likelihood of unsuccessful axillary lymph node staging.

Several clinical studies and meta-analyses support the preference for the ROLL technique as a simple and convenient technique [21]. According to some studies ROLL allows for cosmetically superior and faster excision and simplifies concurrent SLN marking [10,11,22]. Since both the tumor and the axillary lymph node are marked with a single injection, patients subjectively perceive the procedure as less painful than wire placement and peri-areolar radioisotope injection.

Nevertheless, the wire localization method remains a valid alternative as lesion identification and clear margin rates are not significantly different from the ROLL technique [12]. According to the latest international recommendations, WGL is primarily recommended for cases involving extensive microcalcifications, radial scars, and complex sclerosing lesions when a concurrent SLN biopsy is not necessary [23,24,25].

According to both the literature and our observations, wire-guided localization (WGL) may offer advantages in certain other cases.

When SLN marking is not required, WGL provides a way to avoid unnecessary exposure to radioactivity. In addition, more successful SLN visualization was achieved with peri-areolar subdermal injection of the radioisotope, particularly in cases where the tumor became non-palpable after neoadjuvant chemotherapy. In contrast, the ROLL technique, when used without peri-areolar radiotracer injection, is more likely to result in unmarked axillary sentinel lymph nodes [26]. This outcome is influenced by factors such as the intra-mammary location of the tumor, breast size, and breast density.

In situations where the SLN is not identified with nuclide injection, an additional intraoperative blue dye may be used, which is also injected into the subareolar lymphatic plexus [27]. If sentinel node marking proves unsuccessful, surgeons are compelled to consider axillary lymph node dissection for nodal staging.

Our randomized study shows that the ROLL technique is superior to WGL, being less painful for the patient and more ergonomic for the medical team. The procedure was faster, consistent with existing literature, but the difference in procedure time, although statistically significant, lacks clinical relevance as it is only a few minutes. No significant differences were seen in surgical times or specimen size, although results for resected tissue size were mixed, with some tests showing significance and others not.

Although the subjective ease of localization of non-palpable breast tumors is facilitated during the ROLL technique, it can be affirmed that the WGL technique with subareolar injection of the radiocolloid is more successful in terms of visualization of the SLN [28,29].

In order to avoid the morbidity associated with extensive axillary surgery, achieving a high rate of accurate SLN identification is crucial. Consequently, it is imperative to recognize the elevated risk of SLN marking failure.

Therefore, it is necessary to determine in which cases ROLL should be avoided/substituted with another localization technique involving peri-areolar radioisotope injection.

Our results show that in large, adipose breasts, particularly when the tumor is located in the inner quadrants, especially the inner-lower quadrant, isotope tracking with the ROLL technique failed to mark sentinel lymph nodes four times more often than peri-areolar isotope injection used with WGL (67% vs. 14%).

Although the number of cases limited the statistical power of the analysis, there seems to be a strong clinical relevance of our observation with respect to the application in inner quadrant lesions.

Larger case studies further support peri-areolar marking over peri-tumoral injection. For instance, a prospective, randomized, multicentric study compared peri-tumoral and peri-areolar injections in 449 patients. The results showed that the detection rate of axillary SLN by lymphoscintigraphy was significantly higher in the peri-areolar group (85.2%) than in the peri-tumoral group (73.2%). Intraoperative blue dye detection rates were similar in both groups in this study, but the peri-areolar group showed higher SLN detection rates with lymphoscintigraphy and probe use, as well as higher ex vivo SLN counts [30].

Accurate SLN identification is crucial to avoid extensive axillary surgery morbidity. Thus, it is essential to identify cases with a higher risk of SLN marking failure and consider alternative localization techniques using peri-areolar radioisotope injection. After NAT, the integration of targeted axillary lymph node biopsy [31] may further improve patient outcomes. By refining surgical techniques and customizing approaches based on individual patient characteristics, such as tumor location and breast density, we can optimize surgical outcomes and minimize unnecessary interventions [32].

In addition to the traditional methods, combined peri-tumoral and peri-areolar isotope marking should be considered for improved sentinel lymph node identification. Utilizing techniques that employ peri-areolar isotope injection for SLN marking may also improve accuracy. Superparamagnetic iron oxide nanoparticles offer a promising alternative as they are less dependent on the injection site, potentially improving sentinel marking outcomes [33]. In addition, indocyanine green, known for its safety and effectiveness, has demonstrated non-inferior results in localizing non-palpable tumors and may enhance SLN marking without exposing patients to harmful radiation [34].

Research is ongoing to evaluate the feasibility and accuracy of these novel techniques. For instance, the injection of carbon nanoparticle suspensions has emerged as a reliable alternative that may enhance targeted axillary dissection procedures in breast cancer patients who achieve a pathologic complete response after NAT [35]. The introduction of such innovative approaches is essential to improve surgical precision and reduce complications associated with traditional localization methods.

These considerations are particularly important for large, adipose breasts with tumors in the inner quadrants, as demonstrated by our results.

The preference for ROLL due to its simplicity, patient comfort, and cosmetic outcomes is well-supported by the literature [6]. However, this study also appropriately acknowledges the continued relevance of wire localization, especially in specific clinical scenarios.

The analysis of SLN marking failures highlights an important area of concern and suggests a tailored approach based on individual patient characteristics, such as breast size, density, and tumor location.

### 4.1. Suggested Protocol for Non-Palpable Breast Tumor Localization

Given the variability in effectiveness between ROLL and WGL, this protocol emphasizes a patient-specific approach to optimize SLN identification and minimize unnecessary axillary surgeries:

#### 4.1.1. Patient Assessment

Evaluate breast size, density, and tumor location (with special attention to inner quadrant tumors).Prioritize WGL or another non-palpable breast tumor localization method using peri-areolar radiocolloid injection for tumors located in the inner quadrants, especially in large, adipose breasts, where ROLL is less likely to achieve adequate SLN marking.

#### 4.1.2. Preferred Technique

ROLL: Use as the primary localization method for non-palpable tumors, given its advantages in patient comfort, procedural simplicity, and cosmetic outcomes.WGL: Prefer for patients with tumors in the inner quadrants, particularly those who have undergone neoadjuvant chemotherapy, as WGL demonstrates superior SLN identification in these cases.

#### 4.1.3. Intraoperative Contingency Plan

If SLN marking with ROLL fails, intraoperative blue dye injection into the subareolar plexus should be used.Surgeons should be prepared to perform axillary lymph node dissection if SLN identification remains unsuccessful.

#### 4.1.4. Post-Chemotherapy Considerations

For patients with tumor regression after neoadjuvant chemotherapy, consider avoiding ROLL when SLN identification is critical, as its success rate without peri-areolar isotope injection in such cases is lower compared to WGL, especially in adipose involution of large breasts.This protocol aims to optimize SLN marking success and reduce axillary surgery morbidity by selecting the appropriate technique based on individual tumor and breast characteristics.

## 5. Conclusions

Patients report less pain with the ROLL procedure compared to wire placement. From both a radiological and surgical perspective, tumor localization is technically easier with ROLL. Although ROLL has advantages in several aspects, there was no significant difference in overall tumor localization accuracy. However, axillary SLN marking is more often inadequate with peritumoral (ROLL) marking, particularly for tumors located in the inner quadrants. In these situations, the wire-guided technique or alternative approaches using peri-areolar isotope injection are recommended to ensure more accurate SLN identification and reduce the risk of unnecessary axillary surgery.

## Figures and Tables

**Figure 1 biomedicines-12-02466-f001:**
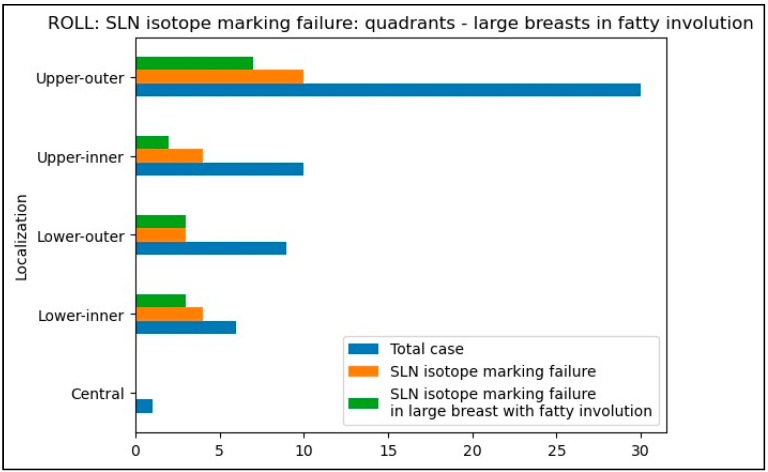
SLN isotope marking failure in the different quadrants and the proportion of large breasts in fatty involution-ROLL group.

**Figure 2 biomedicines-12-02466-f002:**
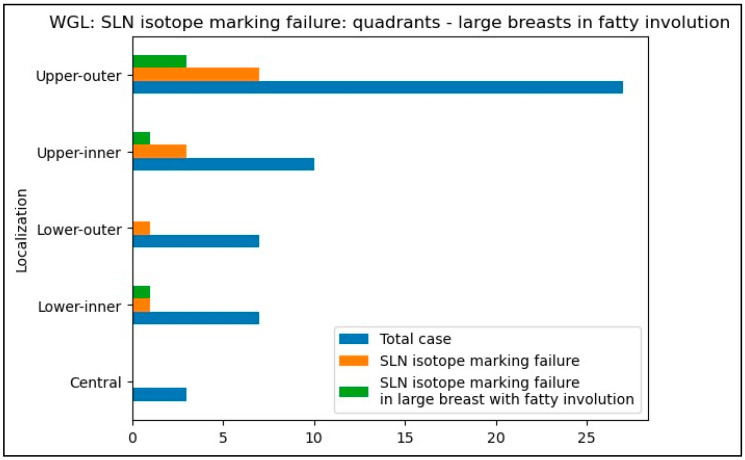
SLN isotope marking failure in the different quadrants and the proportion of large breasts in fatty involution-WGL group.

**Figure 3 biomedicines-12-02466-f003:**
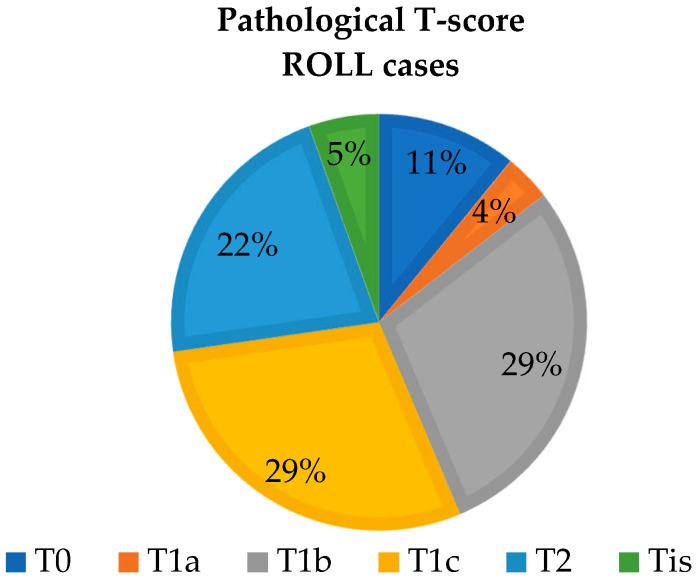
The primary tumor size distribution based on postoperative histopathological examination of the excised specimen in the ROLL group.

**Figure 4 biomedicines-12-02466-f004:**
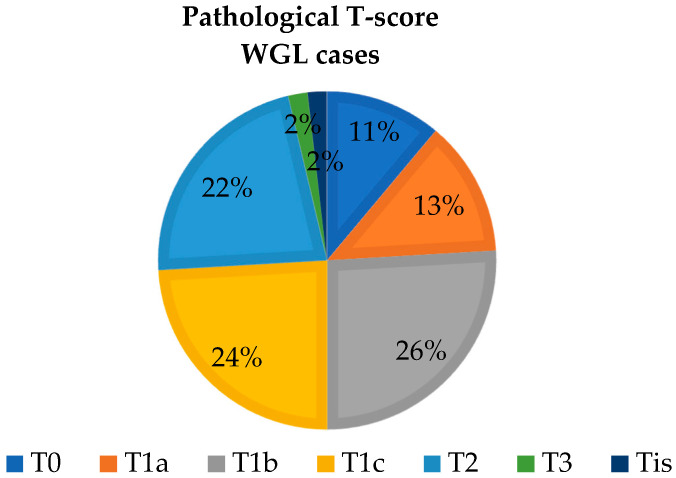
The primary tumor size distribution based on postoperative histopathological examination of the excised specimen in the WGL group.

**Table 1 biomedicines-12-02466-t001:** The ratio of cases of non-palpable lesions in primary surgeries and operations after NAT.

ROLL/WGL Group	Cancer Screening	Neoadjuvant Therapy
ROLL group	43 (77%)	13 (23%)
WGL group	44 (79%)	10 (18%)

**Table 2 biomedicines-12-02466-t002:** Breast size and type distribution in the ROLL group (*n* = 56).

Type and Size of Breast in the ROLL Group (*n* = 56)	Glandular	Adipose	Retro-Mamillary Fibrosis	Adenotic	Fibrotic	Total
Small	8 (14%)	1 (2%)	0	3 (5%)	1 (2%)	13 (23%)
Medium	9 (16%)	2 (4%)	1 (2%)	0	0	12 (21%)
Large	10 (18%)	20 (36%)	0	0	1 (2%)	31 (55%)

**Table 3 biomedicines-12-02466-t003:** Breast size and type distribution in the WGL group (*n* = 54).

Type and Size of Breast in the WGL Group (*n* = 56)	Glandular	Adipose	Retro-Mamillary Fibrosis	Adenotic	Fibrotic	Total
Small	7 (13%)	2 (4%)	4 (7%)	0	0	13 (24%)
Medium	13 (24%)	4 (7%)	4 (7%)	1 (2%)	0	22 (40%)
Large	8 (15%)	10 (19%)	0	1 (2%)	0	19 (35%)

**Table 4 biomedicines-12-02466-t004:** Distribution of tumor localization in breast quadrants.

Localization *	ROLL(*n* = 56)			WGL (*n* = 54)		
	Laterality		Number of Cases	Laterality		Number of Cases
	Left	Right		Left	Right	
Lower-inner q.	3 (5%)	3 (5%)	6 (11%)	4 (7%)	3 (5%)	7 (13%)
Upper-inner q.	8 (14%)	2 (4%)	10 (18%)	8 (14%)	2 (4%)	10 (18%)
Central	1 (2%)	0	1 (2%)	1 (2%)	2 (4%)	3 (5%)
Lower-outer q.	1 (2%)	8 (14%)	9 (16%)	2 (4%)	5 (9%)	7 (13%)
Upper-outer q.	14 (25%)	16 (29%)	30 (54%)	11 (20%)	16 (29%)	27 (48%)

* Provides detailed data on the distribution of tumors across different quadrants of the breast in both the ROLL and WGL groups, considering both the left and right sides. The majority of tumors were located in the upper-outer quadrant for both groups, reflecting a common site for breast tumors.

**Table 5 biomedicines-12-02466-t005:** SLN marking failure.

	ROLL (*n* = 56)		WGL (*n* = 54)	
Localization	Isotope Marking Failure	Patent Blue Marking Failure	Isotope Marking Failure	Patent Blue Marking Failure
Lower-inner q.	4 (67%)	4 (67%)	1 (14%)	2 (29%)
Upper-inner q.	4 (40%)	6 (60%)	1 (10%)	0
Central	0	0	0	1 (33%)
Lower-outer q.	3 (33%)	1 (11%)	0	0
Upper-outer q.	9 (30%)	11 (37%)	2 (7%)	6 (22%)

**Table 6 biomedicines-12-02466-t006:** Comparison of SLN marking failure rates.

Localization	Isotope Marking Failure	Patent Blue Marking Failure
Lower-inner quadrant	0.0265 *	0.0847
Upper-inner quadrant	0.0607	0.0017 **
Central	-	-
Lower-outer quadrant	0.0451 *	0.1812
Upper-outer quadrant	0.0155 *	0.1169

*p* values: * *p* < 0.05, ** *p* < 0.01.

**Table 7 biomedicines-12-02466-t007:** Rate and percentage of SLN isotope marking depending on the Breast Size and Type Distribution in the ROLL group (*n* = 56).

Type and Size of Breast in the ROLL Group (*n* = 56)	Glandular	Adipose	Retro-Mamillary Fibrosis	Adenotic	Fibrotic	Total
Small	8/8 (100%)	1/1(1)	0/0 (0)	3/3 (1)	1/1 (1)	13/13 (1)
Medium	9/8 (89%)	2/1 (50%)	1/1 (100%)	0/0	0/0 (0)	12/10 (83%)
Large	10/6 (60%)	20/5 (25%)	0/0	0/0	1/1 (100%)	31/12 (39%)

**Table 8 biomedicines-12-02466-t008:** Rate and percentage of isotope marking depending on the breast size and type distribution in the WGL group (*n* = 54).

Type and Size of Breast in the WGL Group (*n* = 54)	Glandular	Adipose	Retro-Mamillary Fibrosis	Adenotic	Fibrotic	Total
Small	7/7 (100%)	2/2 (100%)	4/3 (75%)	0/0	0/0	13/12 (92%)
Medium	13/9 (69%)	4/4 (100%)	4/4 (100%)	1/1 (100%)	0/0	22/18 (82%)
Large	8/6 (75%)	10/5 (50%)	0/0	1/1 (100%)	0/0	19/12 (63%)

## Data Availability

Data are contained within the article and Appendix A.

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
