# Peer review of "Non-Palpable Breast Cancer: A Targeting Challenge–Comparison of Radio-Guided vs. Wire-Guided Localization Techniques"

_biomedicines, 2024, doi:10.3390/biomedicines12112466_

Round 1

Reviewer 1 Report

Comments and Suggestions for Authors

This manuscript presents a new and well-planned comparison of the ROLL and WGL techniques for marking both nonpalpable breast tumors as well as sentinel lymph nodes. Thorough work-up with minute analysis of efficacy, patient comfort, and associated radiological outcomes related to both techniques makes it very systematic and comprehensive. The results confirm that though ROLL has superiorities concerning patient comfort, simplicity of procedure, and cosmetic outcome, WGL is superior in the accurate marking of SLN, especially in cases with inner quadrants or large and adipose breasts.

Critique and Suggested Improvements:

- Sample Size and Statistical Power: The main problem/limitation of the present study is the few samples which have been said to decrease its statistical power, particularly pertaining to the sentinel lymph node marking failure analysis. This limitation should, in turn, be more clearly brought out during the discussion by its authors and its results replicated in larger studies in support of its conclusion. Please mention similar up-to-date studies to make comparison of sample size and results.

- Up-to-date literature: The authors need to update their Introduction/Discussion sections. Many references are too old! Stress on the importance of pre-surgical localization techniques also highlighting the superiority of oncological outcomes in patients treated with breast conserving surgery after neoadjuvant therapy. Cite this research: PMID: 39362047 to improve the quality of your manuscript.

- Failure Rate for Sentinel Lymph Node Marking: The manuscript emphasizes the higher failure rate of SLN marking in the ROLL group, especially for inner quadrant tumors. The discussion does not dwell on possible solutions or alternative techniques that might help to overcome this problem. The authors could expand on possible strategies to improve SLN marking with ROLL, such as combining it with peri-areolar isotope injection, or suggesting further research into enhancing this technique for inner quadrant tumors.

Author Response

Dear Reviewer,

Thank you for your valuable feedback. We have carefully considered your suggestions and have made the necessary revisions to the manuscript accordingly.

  1. Sample Size and Statistical Power:

Reviewer’s Comment: The main limitation of the study is the small sample size, which decreases its statistical power, especially for the sentinel lymph node (SLN) marking failure analysis. This limitation should be clearly discussed, and similar studies with comparable sample sizes should be mentioned.

Response: The limitation of the small sample size and its impact on statistical power is now acknowledged, particularly with regard to the analysis of SLN marking failure. The discussion has been expanded to emphasize this limitation and stress the need for future studies with larger sample sizes to validate our findings. Additionally, relevant studies with larger sample sizes have been referenced to provide contextual support and substantiate our conclusions.

Changes Made:

  • Expanded the discussion to explicitly acknowledge the limitation of small sample size.
  • Included references to larger studies for comparison, particularly in relation to SLN marking failure rates.
  1. Up-to-date literature:

Reviewer’s Comment: The authors need to update their Introduction/Discussion sections. Many references are outdated. Also, highlight the importance of pre-surgical localization techniques, including the superior oncological outcomes in patients treated with breast-conserving surgery after neoadjuvant therapy. Please include the study: PMID: 39362047.

Response: The Introduction and Discussion sections have been updated with more recent references. In particular, new literature has been incorporated on the advances in neoadjuvant therapy (NAT) and the role of pre-surgical localization techniques. Furthermore, we have cited the recommended study (PMID: 39362047), which emphasizes the benefits of breast-conserving surgery post-NAT. This addition has reinforced the discussion on the oncological outcomes associated with these approaches.

It should be noted that while the majority of the references have been updated to reflect the latest research, a few older references have been retained due to their historical significance. For instance, these references pertain to the first use of various localization techniques, which are essential for understanding the evolution of these methods over time.

Changes Made:

  • Updated the Introduction and Discussion sections with recent studies.
  • Incorporated the suggested study (PMID: 39362047) and enhanced the discussion of breast-conserving surgery after NAT.
  1. Failure Rate for Sentinel Lymph Node Marking:

Reviewer’s Comment: The discussion could expand on possible strategies to improve SLN marking in the ROLL group, especially for inner quadrant tumors, and mention alternative techniques such as peri-areolar isotope injection.

Response: The discussion has been expanded to include strategies for improving SLN marking in the ROLL group, especially for challenging cases like inner quadrant tumors. We have highlighted the use of peri-areolar isotope injection as a potential method to improve SLN identification. Furthermore, the need for further research into the combination of techniques, such as ROLL with peri-areolar injection, to enhance accuracy was highlighted.

Changes Made:

  • Expanded the discussion to address potential strategies, including peri-areolar isotope injection, to improve SLN marking in the ROLL group.
  • Suggested directions for future research in enhancing ROLL accuracy for challenging tumor locations.
  1. Larger Case Studies and Newer Localization Techniques:

Reviewer’s Comment (implied in feedback): Consider discussing larger studies and newer localization techniques to further support your findings.

Response: In the revised discussion, a section has been added that refers to larger case studies supporting the use of peri-areolar marking over peri-tumoral injection. In addition, we explored emerging technologies such as superparamagnetic iron oxide nanoparticles, indocyanine green, and carbon nanoparticle suspension injection as promising alternatives to traditional methods. These advancements have the potential to improve SLN marking and surgical outcomes, particularly in challenging cases.

Changes Made:

  • Added a discussion of larger case studies supporting peri-areolar marking over peri-tumoral injection.
  • Discussed newer localization techniques and their potential benefits, including superparamagnetic iron oxide nanoparticles, indocyanine green, and carbon nanoparticle suspension injection.

We trust that these revisions have addressed your concerns and improved the quality of the manuscript. Thank you again for your insightful comments, which have helped to refine and reinforce the quality of our work. All changes have been indicated by means of highlighting for your review.

Reviewer 2 Report

Comments and Suggestions for Authors

The manuscript entitled " NON-PALPABLE BREAST CANCER: A TARGETING CHALLENGE - COMPARISON OF RADIO-GUIDED VS WIRE-GUIDED LOCALIZATION TECHNIQUES" discusses the outcome of the research that investigates the impact of using radio-guided occult lesion localization (ROLL) technique versus wire-guided localization (WGL).

They concluded ROLL provided advantages in terms of patient comfort, and logistical simplicity, while WGL was superior for axillary sentinel lymph node marking.

The research conduct is very good, and the study design is well-prepared. The abstract presented the outcomes of the research. The introduction gave the required information. The results are clear and well presented in nice figures.

The conclusion part reflects the research outcomes.

 In general, this research has a scientific essence for certain readers. However, I have some concerns:

1.      There are a lot of previous works with similar conclusions, therefore, the novelty of this paper might be affected.

2.      P value expression is very odd, for instance (WGL (3.78 ±2.03) p=0.0000000007)) preferred to use appropriate significant decimals.

3.      Minor English language proofreading is required.

Author Response

Dear Reviewer,

Thank you for your valuable feedback and comments on our manuscript. We have carefully considered your suggestions and made revisions to address your concerns. All changes have been highlighted in the revised manuscript for your convenience.

  1. Novelty of the Paper:

Reviewer’s Comment: There are a lot of previous works with similar conclusions, therefore, the novelty of this paper might be affected.

Response: We acknowledge your concern regarding the novelty of our work. While similar studies have been conducted, we believe our study offers novel insights by specifically focusing on the comparison between ROLL and WGL in the context of sentinel lymph node (SLN) marking, which has not been extensively addressed in randomized studies. Moreover, we provide an in-depth evaluation of patient comfort, surgeon ergonomics, and radiological outcomes, which adds value to the existing body of research. Additionally, we discuss the importance of advancing the current techniques by integrating newer methods, which have been highlighted in our revised discussion.

Changes Made:

  • Clarified the novel aspects of our study in the introduction and discussion sections, particularly regarding the focus on SLN marking and the evaluation of newer localization techniques.
  1. P Value Expression:

Reviewer’s Comment: P value expression is very odd, for instance, (WGL (3.78 ±2.03) p=0.0000000007)). It is preferred to use appropriate significant decimals.

Response: We agree that the expression of p-values needed adjustment for the purposes of clarity and consistency. The manuscript has been revised to present the p-values rounded to four decimal places, in line with standard reporting conventions.

Changes Made:

  • Revised the p-values throughout the manuscript to ensure they are rounded to four decimal places and consistently expressed.
  1. English Language Proofreading:

Reviewer’s Comment: Minor English language proofreading is required.

Response: The manuscript has been thoroughly reviewed and refined by a professional English translator to ensure the highest language quality. Minor grammatical issues and unclear phrasing have been corrected to improve clarity and readability.

Changes Made:

  • Conducted proofreading by a professional and made corrections to improve the overall language and readability of the manuscript.

We hope that these revisions will address your concerns and improve the overall quality of the manuscript. Thank you again for your thoughtful comments, which have helped to refine our work.

Round 2

Reviewer 1 Report

Comments and Suggestions for Authors

The manuscript can be accepted in the present form